# Cognitive Vergence Recorded with a Webcam-Based Eye-Tracker during an Oddball Task in an Elderly Population

**DOI:** 10.3390/s24030888

**Published:** 2024-01-30

**Authors:** August Romeo, Oleksii Leonovych, Maria Solé Puig, Hans Supèr

**Affiliations:** 1Department Cognition, Development and Educational Psychology, University of Barcelona, 08035 Barcelona, Spain; 2Bioinformatics and Biomedical Signals Laboratory, Polytechnical University of Catalonia, 08028 Barcelona, Spain; 3Assesment Unit of Cognition and Attention in Learning, Psychology Clinic, 08035 Barcelona, Spain; mariasoleacap@gmail.com; 4Braingaze S.L., 08302 Mataró, Spain; 5Institute of Neurosciences (UBNeuro), University of Barcelona, 08035 Barcelona, Spain; 6Institut de Recerca Sant Joan de Déu (IRSJD), 08950 Barcelona, Spain; 7Catalan Institution for Research and Advanced Studies (ICREA), 08010 Barcelona, Spain

**Keywords:** eye tracking, iris, vergence eye movement, Alzheimer, MCI, detection

## Abstract

(1) Background: Our previous research provides evidence that vergence eye movements may significantly influence cognitive processing and could serve as a reliable measure of cognitive issues. The rise of consumer-grade eye tracking technology, which uses sophisticated imaging techniques in the visible light spectrum to determine gaze position, is noteworthy. In our study, we explored the feasibility of using webcam-based eye tracking to monitor the vergence eye movements of patients with Mild Cognitive Impairment (MCI) during a visual oddball paradigm. (2) Methods: We simultaneously recorded eye positions using a remote infrared-based pupil eye tracker. (3) Results: Both tracking methods effectively captured vergence eye movements and demonstrated robust cognitive vergence responses, where participants exhibited larger vergence eye movement amplitudes in response to targets versus distractors. (4) Conclusions: In summary, the use of a consumer-grade webcam to record cognitive vergence shows potential. This method could lay the groundwork for future research aimed at creating an affordable screening tool for mental health care.

## 1. Introduction

Our eyes, constantly in motion, play a pivotal role in visual information processing. Even when our gaze is steady, tiny eye movements, known as fixational or micro saccades, are crucial. These movements not only prevent the loss of conscious vision [1], but also aid in attention shifts [2,3], enhance visual sensitivity [4,5], and improve visual acuity [6,7].

Vergence, another form of eye movement [8,9,10,11,12], involves the eyes moving in opposite directions to achieve and maintain monocular vision. Our research has discovered a new role for vergence eye movements in cognitive processing. We observed that the eyes briefly converge following the presentation of a visual stimulus [13]. These vergence responses are more pronounced when the stimulus is attended, perceived, or retained in memory (e.g., see [13,14,15]). This indicates a potential role of vergence in attention. Additional evidence comes from observations that individuals with attentional difficulties exhibit poor vergence responses during an attentional task [16]. We refer to this phenomenon as cognitive vergence. However, some authors suggest that vergence estimates from infrared eye trackers represent both rotation of the eye and pupil dynamics [17].

Cognitive vergence responses appear early and increase as the processing of a stimulus reaches a level where their strength correlates with behavioral performance. This suggests that vergence responses could predict the extent to which a stimulus is processed. Therefore, measuring cognitive vergence could potentially serve as an objective marker for detecting cognitive problems. Indeed, AI classifier models using cognitive vergence responses as input have successfully identified patients with ADHD [16] and Mild Cognitive Impairment (MCI) [18]. In patients with MCI, attended stimuli are accompanied by a weak enhancement, whereas Alzheimer patients show no difference in vergence responses to attended and unattended stimuli. Such models can even predict the risk of patients with MCI developing Alzheimer’s disease [19].

Eye gaze tracking is typically performed using specially designed devices that employ infrared light to detect pupil size and center and estimate gaze position. This technology often involves relatively expensive devices, limiting its widespread adoption and primarily confining its use to research and assistive applications. However, more applications are being developed, such as those for artistic purposes [20] and human–robot interfaces [21,22]. Additionally, new methods are emerging [23]. Some of them utilize advanced imaging techniques in the visible light spectrum to estimate gaze position using the iris of the eye [24]. This advancement paves the way for developing consumer-grade eye tracking technology that could potentially be used to detect mental health conditions by measuring cognitive vergence. In this study, we explored the feasibility of such a technique by testing patients with MCI. Participants performed a brief computerized visual oddball paradigm while cognitive vergence eye movements were measured from images recorded by a webcam. Eye positions were also recorded simultaneously with a remote infrared-based pupil eye tracker.

Our results indicate that a differential vergence response to the oddball task stimuli (targets and distractors) can be measured with both a webcam-based iris tracker and infrared pupil tracker. This signifies that vergence estimates captured from infrared and webcam-based eye trackers reflect eye rotation to a greater extent than they reflect a misinterpretation of pupil diameter changes. Although the absolute magnitude of the vergence angle varied between trackers, the modulation pattern and index of the vergence responses were similar for both trackers. The findings imply that employing a consumer-grade webcam could be a viable method for capturing cognitive vergence. This holds promise for future research aimed at creating an affordable screening instrument for mental health care.

## 2. Materials and Methods

### 2.1. Subjects

We conducted our study with participants recruited from a private day care center for the elderly in Barcelona. The clinical professional at the care center extended invitations to volunteers, and a total of 28 subjects (9 men and 19 women; mean (SD) age: 70.3 (6.8) years) willingly participated in the study. The Montreal Cognitive Assessment (MoCA) was administered to all participants to evaluate their cognitive abilities. The inclusion criterion was set based on MoCA score ranging between 18 and 28 out of a possible 30 points.

The exclusion criteria were as follows: (1) history of neurological disease with clinically relevant impact on cognition (e.g., cerebrovascular disease); (2) severe psychiatric disorder; (3) presence of relevant visual problems; and (4) problems for understanding spoken or written Spanish language. 

### 2.2. Ethical Statement

Participants and their relatives received detailed instructions for the experiments. Prior to enrollment, patients or family members signed a written informed consent for their participation in accordance with the Declaration of Helsinki. This study was approved by the ethics committees of the University of Barcelona.

### 2.3. Apparatus

We used the BGaze software (version 1.17.2; Braingaze SL, Mataró, Spain) on a laptop (MSI CX62 6QD) to present the visual stimuli and record eye position data. The faces of the participants were recorded with the integrated webcam (HD type, 30 fps, 720 p) while performing the task. We chose a webcam of standard quality based on our pilot testing, which demonstrated satisfactory results. The resolution of the screen (HD 15.6”) was 1366 × 768 pixels and the remote eye tracker (ET) used was an X2-30 (30 Hz, Tobii Technology AB, Stockholm, Sweden).

### 2.4. Experimental Procedure 

The task was performed in the living room of each patient’s home in order to have conditions in an operational setting ensuring sufficient ambient light without reflecting light sources. However, due to variations in lighting in each room, standard conditions could not be established (Figure 1B). The laptop was positioned on a table with the screen slightly inclined so that the entire face was captured by the webcam. The subjects were seated approximately 50 cm from the screen on which the stimuli were presented. No chin rest was used so patients could freely move their heads, and they were allowed to wear corrective lenses.

### 2.5. Paradigm

The experiment employed a visual oddball paradigm, comprising a sequence of 100 trials. Each trial began with a gray screen (mask) displayed for 2000 ms, followed by a centrally presented visual stimulus for an equal duration (Figure 1). This stimulus consisted of an 11-character string of letters, randomly selected and varying in case. To avoid bias, these strings did not form acronyms or meaningful words. The strings were identical except for their color. In 80% of the trials, all characters were blue, while in the remaining 20%, they were red. Participants were instructed to focus on the screen and press a key only when the characters were red. Thus, red character strings served as targets, and blue ones as distractors. The stimuli were presented randomly. The task, lasting approximately 6 min, involved recording pupil positions using a remote eye tracker and capturing the participant’s face with a webcam.

### 2.6. Webcam-Based Eye Tracking (WC)

To obtain cognitive vergence measurements from the webcam images, we used the model described in [25], which captures the 3D head poses, facial expression deformations, and 3D eye gaze states using a single RGB camera. The whole system consists of several components. First, important facial features are automatically detected and tracked, and the optical flow of each pixel in the face region is computed. Then, a data-driven 3D facial reconstruction technique is performed to reconstruct the 3D head pose and large-scale expression deformations using multi-linear expression deformation models. A pixel classifier then automatically annotates the iris and pupil pixels in the eye region, which is bounded by detected facial landmarks in the eye region. Additionally, the outer contour of the iris (i.e., the limbus) is extracted to further improve the robustness and accuracy of the gaze tracker. A DCNN-based segmentation method is used to perform a frame-by-frame pixel extraction of the iris including the pupil region. The convolutional neural network is used to predict the probability that each pixel belongs to the entire iris, including the pupil region. To track the gaze states in the video sequences, the geometric shape and 3D position of the eyeballs and the radius of the iris region together with the limbus are estimated.

### 2.7. Cognitive Vergence Calculation

Data points from the infrared eye tracker that did not correspond to valid pupil detections (i.e., whenever the validity score given by the eye tracker software had a non-zero value) were marked out. Trials with too many invalid data points (15 points or more) were discarded. The exclusion rate was 33%. Finally, interpolation was used to create sequences of evenly spaced points. In the case of the webcam eye tracker, all trials were included in the data analysis. 

To calculate vergence changes, we transformed the coordinates of the left and right eye provided by the eye tracker into angular values. Rather than the vergence angle itself, *γ* (for example), we focus on the relative vergence modulation $V(t)\equiv \frac{\gamma (t)-\gamma_{o}}{\max\left|\gamma (t)-\gamma_{0}\right|}$, where *γ*_0_ represents the γ value at stimulus onset, and the indicated maximum was taken for all absolute values of the difference *γ*(*t*) − *γ*_0_ in the examined time window of each trial. The subtraction of the initial values from each response served the purpose of obtaining relative changes. Subsequently, all V(t) sequences coming from trials in the same condition (target, distractor) were averaged to obtain ‘mean V(t)’ curves.

### 2.8. Data and Statistical Analysis

The peak vergence response was evaluated as the mean in the 400–433 ms window. Delayed responses were calculated as the average response strength over the window 600–1250 ms after stimulus onset. Modulation indices were calculated as mi = (T − D)/(T + D), where T(D) is the mean of the window-averaged vergence responses for all target (distractor) trials. For both tracker methods, the window limits were 300–600 ms. For statistical analysis, we performed a series of comparisons based on the two-tailed *t*-test of all accepted trials or subjects.

## 3. Results

MOCA scores ranged from 11 to 25 (mean ± std: 16.8 ± 3.9) out of a possible 30, indicating that subjects had cognitive impairment. Three subjects were excluded from further analysis of infrared eye tracker data because they did not provide valid pupil recordings. In total, there were 1512 distractor trials and 350 target trials. Removing the same three participants from the webcam data did not significantly change the results. Therefore, we decided to include all 28 subjects in the analysis of the webcam data. The total number of distractor trials was 2240 and there were 560 target trials, but only 2218 and 551 were correctly recorded, respectively.

### 3.1. Cognitive Vergence Responses 

Iris positions were extracted from the webcam images to calculate vergence responses separately for the target and distractor conditions. The average target response of all participants across trials shows a clear increase in vergence angle starting about 300 ms after stimulus onset and peak at about 450 ms, followed by a delay response (Figure 2). The average peak response to targets (mean ± std: 0.036 ± 0.107) was stronger than the initial—i.e., 0–200 ms—averaged vergence responses (mean ± std: −0.002 ± 0.060, *t*-test result *t* = 1.60, *p* = 0.12). The average target delay response (mean ± std: 0.021 ± 0.083) was similar to the initial response strength (*t* = 1.19, *p* = 0.24). Vergence eye movements during distractor trials showed neither a peak response (mean ± std: −0.013 ± 0.062) nor a clear delay response (mean ± std: 0.001 ± 0.056), but a slightly significant response increase of 3.3 × 10^−5^ deg/ms was visible. See also Figure 3 and Table 1.

These findings suggest that webcam-based eye tracking can be used to assess cognitive function in individuals with MCI by measuring vergence eye movements during a visual oddball paradigm.

### 3.2. Infrared Eye Tracker (ET)

Simultaneously with the webcam recording, we recorded vergence responses with a remote infrared-based eye tracker, allowing us to compare both methods. The average vergence response to targets recorded with the infrared eye tracker showed a peak response around 500 ms followed by a delay response (Figure 2B). The results show that 52% of the subjects had a stronger peak response to targets than to distractors. See Figure 4. The average peak response to targets (mean ± std: 0.076 ± 0.534) was significantly (*t*-test result *t* = −2.24, *p* = 0.03) stronger than the average response to distractors (mean ± std: 0.005 ± 0.530). The delay response in the distractor condition showed a strong increase starting at about 600 ms and reached a maximum at 1100 ms.

### 3.3. Comparison of Webcam-Based versus Infrared Eye Tracking

The average curves for initial and peak responses do not significantly differ (*t*-test: *t* = 0.620 *p* = 0.535 and *t* = −0.311 *p* = 0.756, respectively) between webcam-based and infrared-based eye tracking but the delayed responses are significantly different (*t* = 3.798, *p* ≈ 1.4 × 10^−4^).

To compare the differential vergence response recorded by the webcam-based eye tracker with that of the infrared-based eye tracker, we plotted the modulation indices per subject. The window for calculating the modulation index was 300–600 ms. The results show that 75.0% and 66.6% of the subjects showed a positive modulation index, i.e., the responses to targets were stronger than those to distractors when recorded with the infrared-based eye tracker and the webcam-based eye tracker, respectively (Figure 5). However, there was a positive modulation in the webcam eye tracker in 28.4% of the participants (N = 6), while there was a negative modulation in the infrared eye tracker. Seven participants showed a positive modulation index in the infrared tracking but a negative one with webcam tracking, and nine participants showed positive modulation in both trackers. The average modulation index (mi) across subjects for infrared-based eye tracking (miET) was 1.06 ± 2.69, while the average modulation index for webcam-based eye tracking (miWC) was 0.66 ± 4.51. However, this difference was not statistically significant (*t* = 0.40, *p* = 0.70), suggesting that both methods can effectively capture the differential vergence response.

## 4. Discussion

In this study, we compared cognitive vergence responses recorded with a webcam-based eye tracker to those captured with a remote infrared-based eye tracker during an oddball task. Participants were simultaneously tracked using a remote infrared-based pupil eye tracker and a consumer-grade webcam. Both tracking methods effectively captured vergence eye movements and demonstrated robust cognitive vergence responses, where participants exhibited larger vergence eye movement amplitudes in response to targets compared to distractors. Although both trackers exhibited a similar temporal pattern of vergence responses, the absolute response amplitudes were smaller when recorded with the webcam-based tracker, particularly during the delay period, possibly due to differences in recording and computation methods. Additionally, the standard deviation of the responses from the webcam eye tracker is larger than that of the infrared eye tracker, indicating less precise responses. This aligns with previous studies, demonstrating that while webcam eye tracking serves as an alternative to infrared eye tracking, its spatial resolution remains inferior [26].

Yet, both the webcam-based eye tracker and the infrared-based eye tracker produced stronger responses to targets than to distractors, which is in agreement with our previous study showing a differential vergence response in an elderly population [18]. This differential response is typically present in cognitively healthy subjects but is reduced or absent in those with cognitive impairment. Given that all participants in our current study had a history of cognitive impairment, as indicated by their MoCA scores, this could explain why some did not exhibit a differential vergence response. We conducted this study in an uncontrolled environment without the use of a chin rest, which may have introduced additional noise into the eye tracking data. The sensitivity of the trackers to noise is unlikely to be identical due to their different signal detection methods. Despite differences in absolute magnitude, both tracking methods yielded similar temporal patterns of cognitive vergence responses and captured a differential response. This indicates that webcam eye tracking technology may be capable of detecting cognitive decline. An early intervention in response to MCI with pharmaceutical treatment, cognitive therapy, or an adoption of a healthy lifestyle may help to prevent or delay the onset of Alzheimer’s disease. However, available biomarker tools are expensive and invasive, and more accessible solutions are needed. In line with previous reports, the assessment of vergence responses could be a potential candidate to consider for further clinical research in developing an objective, low-cost marker tool for consumers to monitor their cognitive health. In out earlier studies [27], we demonstrated that cognitive vergence is disrupted in individuals with neurodevelopmental disorders such as ADHD and ASD. Consequently, our technology holds promise for potential applications, including the detection of these disorders.

### Cognitive Vergence and Pupil Responses?

The neural mechanisms that govern vergence and pupil size share some overlap, leading to a situation where a vergence eye movement can elicit a pupil response [28]. This interplay results in a complex behavioral relationship (see [13]). Infrared-based eye trackers estimate gaze position using pupil size, center, and corneal reflectance. Some suggest that these metrics may introduce errors in estimating eye movement amplitude [17,29,30,31]. They argue that cognitive vergence may represent pupil dynamics in addition to actual vergence movement [17]. However, other studies indicate that infrared-based tracking is comparable to the search coil method for measuring small fixational eye movements [32]. They also suggest that pupil-related errors may become negligible at viewing distances beyond 50 cm [27], which aligns with the distance we used in our study.

Our study employed a webcam-based eye tracker that estimates gaze by detecting the iris area and limbus. These measurements are independent of pupil detection and remain unaffected by changes in pupil size. We obtained a clear differential vergence response with the webcam-based eye tracker. This lends further support to the notion that cognitive vergence results from the rotation of the eyeball and artifact or error in measurements of pupil size or corneal reflection may not be as influential in determining vergence estimates, as previously suggested.

## 5. Conclusions

In conclusion, our findings suggest that a consumer-grade webcam holds promise as a potential tool for recording cognitive vergence. To establish it as an affordable screening aid, further research is required to validate its clinical effectiveness and demonstrate its applicability in the realm of mental health care. Potential future applications could involve the development of a consumer-oriented tool for regularly monitoring mental health conditions. 

## 6. Patents

The IP of the method to detect cognitive disorders is protected with a patent.

## Figures and Tables

**Figure 1 sensors-24-00888-f001:**
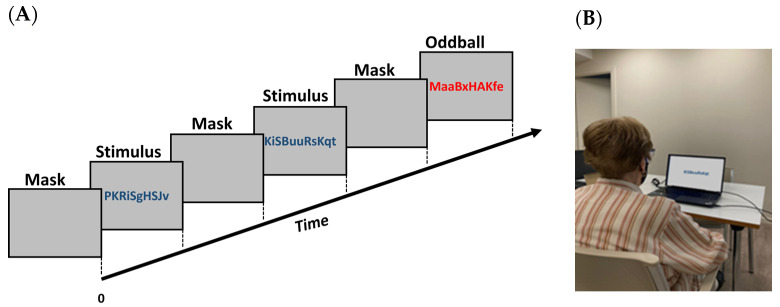
Schematic representation of the oddball task (**A**) and an example of the set-up (**B**). A series of letters is presented for two seconds. In 80% of the trials, the letters were in blue color, and in the remaining 20% (oddball), the letters were in red color.

**Figure 2 sensors-24-00888-f002:**
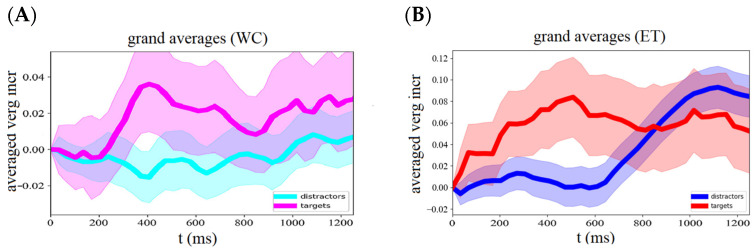
Average vergence eye responses to target and distractor stimuli of an oddball paradigm recorded with the webcam (WC, (**A**)) and IR (ET, (**B**)) eye tracker. Responses to targets are depicted in magenta and red traces and to distractors in (light) blue traces.

**Figure 3 sensors-24-00888-f003:**
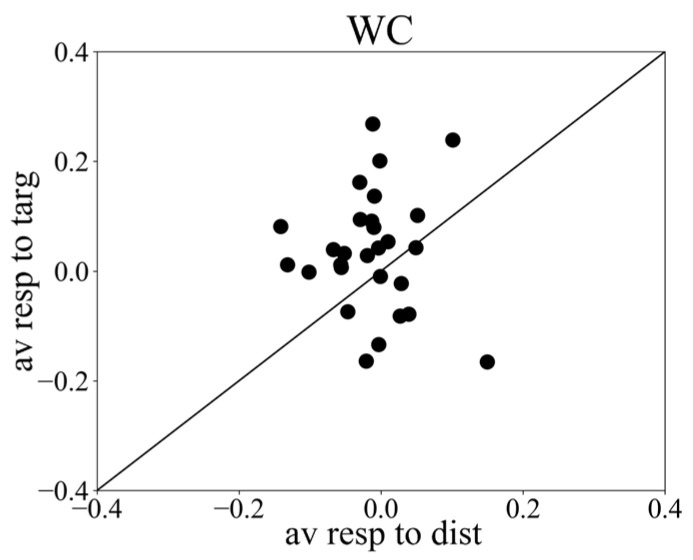
Scatter plot showing peak responses to targets versus peak responses to distractor, from webcam data. *X* axis: averaged vergence response to distractors; *y* axis: averaged vergence response to targets.

**Figure 4 sensors-24-00888-f004:**
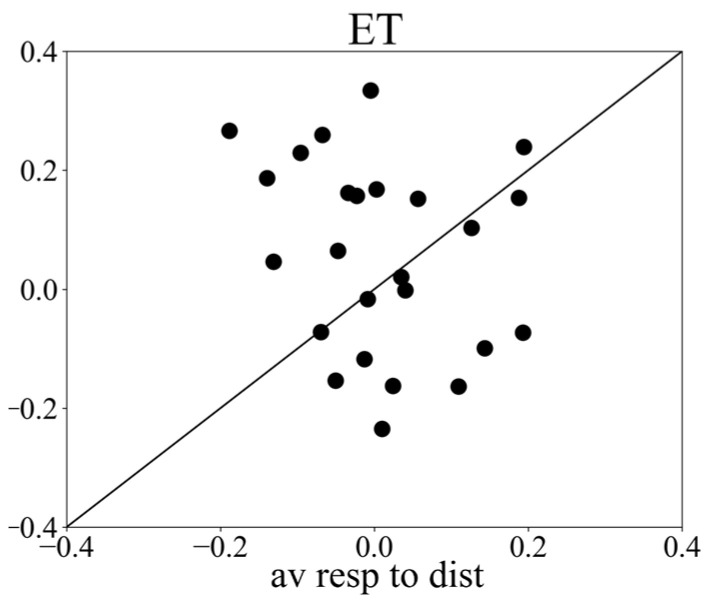
Scatter plot of peak responses, recorded with the remote infrared eye tracker, to targets versus distractors (*x* axis: averaged vergence response to distractors; *y* axis: averaged vergence response to targets).

**Figure 5 sensors-24-00888-f005:**
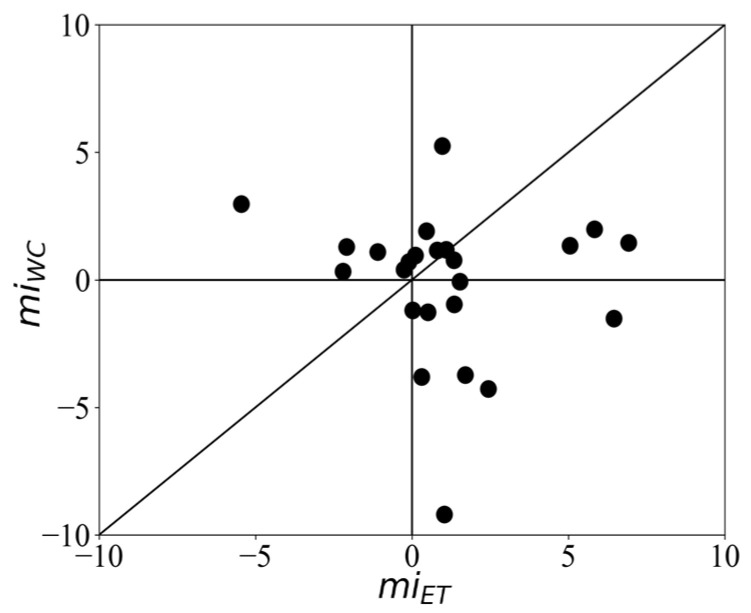
Scatter plot showing the modulation indices of vergence responses recorded by the webcam-based eye tracker (mi_WC_) compared to those recorded by the infrared-based eye tracker (mi_ET_), per subject.

**Table 1 sensors-24-00888-t001:** Table summarizing the obtained results with the webcam eye tracker.

	Initial Response	Peak Response	Delay Response
Target	−0.002 ± 0.060	0.036 ± 0.107	0.021 ± 0.083
Distractor	−0.004 ± 0.044	−0.013 ± 0.062	0.001 ± 0.056

## Data Availability

Data within this study can be requested from H.S. via email.

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
