# Peer review of "Cognitive Vergence Recorded with a Webcam-Based Eye-Tracker during an Oddball Task in an Elderly Population"

_sensors, 2024, doi:10.3390/s24030888_

Round 1

Reviewer 1 Report

Comments and Suggestions for Authors

The authors explored the feasibility of using webcam-based eye tracking to monitor the vergence eye movements of patients with Mild Cognitive Impairment during a visual oddball paradigm.

The results show that a consumer-grade webcam can be a potential tool for recording cognitive vergence , although further research is required to validate its clinical effectiveness and demonstrate its applicability in the realm of mental health care.

The paper is well structured but the presentation of both material and methods and results can be improved. In the following some suggestions to be considered for the paper revision are reported:

-          Section 2.1: no information regarding inclusion and exclusion criteria is provided.

-          Section 2.3: no information regarding the considered webcam selection criteria is provided.  Is the selected webcam the one with the minimum required specs? The selection of two different webcams could have been interesting for an additional comparison.

-          Section 2.4: the lighting condition of one living room can be different from another. More details on the settings are appreciated (lighting condition, webcam position, etc.). Moreover, additional information regarding how different factors can affect the measurements can be useful for the reader to better understand the study.

-          to better compare the results, the reported graphics of WC and ET should be located side by side. The y values seem to be different. Why?

-          Figure 2: y-axis label is missing.

-          A table that summarizes the obtained results should be added, to better highlights the achievements.

-          Discussion shall be extended.

-          Conclusion shall be extended. Some future investigations may be included.

-          Section 4.1 location shall be re-considered. This information may be integrated in the material and methods section.

-          some formulas formatting issues should be fixed.

-          The references are not well formatted, and the inclusion of additional recent references would be appreciated.

Author Response

First and foremost, I would like to express my gratitude to the reviewers for dedicating their time and effort to evaluate the manuscript and provide valuable suggestions for its improvement. Below, I present a point-to-point response

Reply to reviewer 1

Section 2.1: no information regarding inclusion and exclusion criteria is provided.

In the revised manuscript we have now mentioned the inclusion and exclusion criteria

Section 2.3: no information regarding the considered webcam selection criteria is provided.  Is the selected webcam the one with the minimum required specs? The selection of two different webcams could have been interesting for an additional comparison.

After conducting pilot testing, we chose a webcam with standard quality, which demonstrated satisfactory results. We mention this in the revised manuscript. While a higher resolution camera could have given superior results, we opted not to test two or more different webcams as recruitment was slow.  

Section 2.4: the lighting condition of one living room can be different from another. More details on the settings are appreciated (lighting condition, webcam position, etc.). Moreover, additional information regarding how different factors can affect the measurements can be useful for the reader to better understand the study.

We have now included additional details on the settings of the experiment. Also we included a new figure (fig 1B) showing an example of the set-up.

To better compare the results, the reported graphics of WC and ET should be located side by side. The y values seem to be different. Why?

In the revised manuscript we, have place both figures side by side to aid comparison. The y values are different because response amplitudes were smaller when recorded with the webcam-based tracker, possibly due to differences in recording and computation methods. In the revised manuscript we comment this and included statistical analysis comparing both response strengths. The initial and peak values are not different but the delay responses are. We discuss this in the revised manuscript.

Figure 2: y-axis label is missing.

We have corrected this

A table that summarizes the obtained results should be added, to better highlights the achievements.

We have included a table (Table 1) summarizing the results.

Discussion shall be extended.

In the revised manuscript we have amplified the discussion.

Conclusion shall be extended. Some future investigations may be included.

In the revised manuscript, we have extended the conclusion

Section 4.1 location shall be re-considered. This information may be integrated in the material and methods section.

We appreciate your concern. The findings presented in this section diverge from the primary focus of the study; instead, they highlight an additional outcome that we find noteworthy for discussion, particularly as it pertains to a broader issue in eye-tracking research. Recognizing its status as a supplementary aspect of the study, we acknowledge that its placement in the Materials and Methods section may create confusion. In response, we have revised the text to seamlessly align with the ongoing discussion, ensuring a more cohesive and understandable presentation.

Some formulas formatting issues should be fixed.

Thank you for noting. We have corrected the mistakes

The references are not well formatted, and the inclusion of additional recent references would be appreciated.

We have re-formatted the  references and added some new ones

Reviewer 2 Report

Comments and Suggestions for Authors

The paper explores the feasibility of using webcam-based eye tracking to monitor the vergence eye movements of patients with Mild Cognitive Impairment (MCI) during a visual oddball paradigm. The paper is interesting and suitable for the journal. However, the following points need to be carefully addressed before publication:

1) The main contributions of the paper should be better clarified with respect to the present literature. 

2) The state of the art should be improved, considering additional works on the use of eye tracking technology in several fields. See for instance:

Comparing two safe distance maintenance algorithms for a gaze-controlled HRI involving users with SSMI. ACM Transactions on Accessible Computing (TACCESS), 15(3), 1-23.

Human–robot interaction through eye tracking for artistic drawing. Robotics, 10(2), 54.

Eye tracking and eye expression decoding based on transparent, flexible and ultra-persistent electrostatic interface. Nature Communications, 14(1), 3315.

Gaze Controlled Safe HRI for Users with SSMI. In 2021 20th International Conference on Advanced Robotics (ICAR) (pp. 913-918). IEEE.

3) The quality of the figures should be improved. Furthermore, an image of an experimental test should be reported in the text.

4) More details about the participants to the text should be added to the paper. 

5) The statistical analysis and the discussion of the results should be improved.

Author Response

First and foremost, I would like to express my gratitude to the reviewers for dedicating their time and effort to evaluate the manuscript and provide valuable suggestions for its improvement. Below, I present a point-to-point response.

Reply to reviewer 2

1) The main contributions of the paper should be better clarified with respect to the present literature. 

In the revised manuscript, we have explained in more detail de main contribution of the paper

2) The state of the art should be improved, considering additional works on the use of eye tracking technology in several fields. See for instance:

We have included the cited references and added the additional studies on the use of eye tracking

3) The quality of the figures should be improved. Furthermore, an image of an experimental test should be reported in the text.

We have improved the quality of the figures and included an image showing an example of the experimental setup

4) More details about the participants to the text should be added to the paper. 

We have added more information about the participants.

5) The statistical analysis and the discussion of the results should be improved.

In the revised manuscript, we improved the description of the statistical analysis, included a table (table 1) and a comparison between the average curves from both eye trackers.  

Reviewer 3 Report

Comments and Suggestions for Authors

1. The study could benefit from a more detailed description of the sample size determination and the recruitment process.

2. Include more in-depth statistical analysis, particularly regarding the comparison between the webcam-based eye tracker and the traditional infrared-based tracker. This will add robustness to the claim that webcam-based tracking is a viable alternative.

3. How does the sensitivity of the webcam-based eye tracker compare to traditional methods in different lighting conditions and with participants of varying age groups?

4. While the study acknowledges potential limitations due to the uncontrolled experimental environment, a more thorough discussion on how this might have impacted the results would be valuable. Additionally, outlining potential future research directions, especially in terms of scaling the technology for wider clinical use, would be insightful.

5. Could you elaborate on the potential applications of this technology in other cognitive disorders beyond MCI?

6. Please remove the numerical elements from the keywords in your manuscript.

Author Response

First and foremost, I would like to express my gratitude to the reviewers for dedicating their time and effort to evaluate the manuscript and provide valuable suggestions for its improvement. Below, I present a point-to-point response

  1. The study could benefit from a more detailed description of the sample size determination and the recruitment process.

We have included more details about the recruitment and sample size

  1. Include more in-depth statistical analysis, particularly regarding the comparison between the webcam-based eye tracker and the traditional infrared-based tracker. This will add robustness to the claim that webcam-based tracking is a viable alternative.

Overall, the vergence values obtained from the webcam are smaller than those recorded with the infra-eye tracker. This difference may be attributed to variations in recording and computation methods. The objective of this study is to investigate whether distinct vergence responses can be accurately recorded with webcam eye tracking.

We nevertheless compared the vergence responses from both eye trackers, and observed that the initial and peak responses did not show significant differences; only the delay responses varied. We have included this finding in the revised manuscript.

  1. How does the sensitivity of the webcam-based eye tracker compare to traditional methods in different lighting conditions and with participants of varying age groups?

Studies show that webcam eye tracking is an affordable and accessible alternative to lab-based infrared eye tracking (Vos et al., 2022). However, the main bottleneck is the relatively low spatial accuracy. In our study, the standard deviation of the responses from the webcam eye tracker is larger than that of the infra-red eye tracker, indicating less precise responses. We have included this information in the revised manuscript.

As we were specifically interested in the elderly with Mild Cognitive Impairment (MCI), as they are at risk of developing Alzheimer's disease, we did not test young subjects.

  1. 4. While the study acknowledges potential limitations due to the uncontrolled experimental environment, a more thorough discussion on how this might have impacted the results would be valuable. Additionally, outlining potential future research directions, especially in terms of scaling the technology for wider clinical use, would be insightful.

We have amplified the discussion with the points you mentioned

  1. Could you elaborate on the potential applications of this technology in other cognitive disorders beyond MCI?

In previous studies, we demonstrated that cognitive vergence is disrupted in individuals with neurodevelopmental disorders such as ADHD and ASD. Consequently, our technology holds promise for potential applications, including the objective detection of these disorders.

We mention this in the revised manuscript.

  1. Please remove the numerical elements from the keywords in your manuscript.

We have removed the numbers.

  1. The study could benefit from a more detailed description of the sample size determination and the recruitment process.

We have included more details about the recruitment and sample size

  1. Include more in-depth statistical analysis, particularly regarding the comparison between the webcam-based eye tracker and the traditional infrared-based tracker. This will add robustness to the claim that webcam-based tracking is a viable alternative.

Overall, the vergence values obtained from the webcam are smaller than those recorded with the infra-eye tracker. This difference may be attributed to variations in recording and computation methods. The objective of this study is to investigate whether distinct vergence responses can be accurately recorded with webcam eye tracking.

We nevertheless compared the vergence responses from both eye trackers, and observed that the initial and peak responses did not show significant differences; only the delay responses varied. We have included this finding in the revised manuscript.

  1. How does the sensitivity of the webcam-based eye tracker compare to traditional methods in different lighting conditions and with participants of varying age groups?

Studies show that webcam eye tracking is an affordable and accessible alternative to lab-based infrared eye tracking (Vos et al., 2022). However, the main bottleneck is the relatively low spatial accuracy. In our study, the standard deviation of the responses from the webcam eye tracker is larger than that of the infra-red eye tracker, indicating less precise responses. We have included this information in the revised manuscript.

As we were specifically interested in the elderly with Mild Cognitive Impairment (MCI), as they are at risk of developing Alzheimer's disease, we did not test young subjects.

  1. 4. While the study acknowledges potential limitations due to the uncontrolled experimental environment, a more thorough discussion on how this might have impacted the results would be valuable. Additionally, outlining potential future research directions, especially in terms of scaling the technology for wider clinical use, would be insightful.

We have amplified the discussion with the points you mentioned

  1. Could you elaborate on the potential applications of this technology in other cognitive disorders beyond MCI?

In previous studies, we demonstrated that cognitive vergence is disrupted in individuals with neurodevelopmental disorders such as ADHD and ASD. Consequently, our technology holds promise for potential applications, including the objective detection of these disorders.

We mention this in the revised manuscript.

  1. Please remove the numerical elements from the keywords in your manuscript.

We have removed the numbers.

Round 2

Reviewer 1 Report

Comments and Suggestions for Authors

The authors revised the paper according to the reviewers' comments, improving the quality of presentation of the study.

Author Response

The reviewer had no comments

Reviewer 2 Report

Comments and Suggestions for Authors

The paper has been somehow improved with respect to the previous version. However, the quality of the images is still low. More in detail: 

- in Figure 1A the resolution is too low;

- in Figure 1B it is difficult to see what appears on the screen;

- in Figure 2, 3, 4 and 5, it is difficult to read the labels. The resolution should be improved in all figures.

 The main results of the paper should be better highlighted and commented.

Author Response

- in Figure 1A the resolution is too low;

We have re-drawn the figure and saved it in a higher resolution.

- in Figure 1B it is difficult to see what appears on the screen;

While we currently lack images of the setup with a patient, we have overlaid an image of the task to provide a clearer view of the screen. The primary objective of this image is to showcase the setup's arrangement. We believe this solution is acceptable for the intended purpose.

 -in Figure 2, 3, 4 and 5, it is difficult to read the labels. The resolution should be improved in all figures.

In the revised manuscript, we have improved the quality of the labeling of these figures

-The main results of the paper should be better highlighted and commented.

In the Abstract and Discussion, we have highlighted and commented the main results, and in the Results we added a conclusion to the relevant sections.

Reviewer 3 Report

Comments and Suggestions for Authors

The authors have addressed the suggestions provided I believe the paper can be accepted for publication in the current state 

Author Response

The reviewer had no comments